# Characteristics Associated with the Dual Behavior of Mask Wearing and Vaccine Acceptance: A Pooled Cross-Sectional Study among Adults in Saskatchewan

**DOI:** 10.3390/ijerph19063202

**Published:** 2022-03-09

**Authors:** Ali Bukhari, Daniel A. Adeyinka, Jessica McCutcheon, Natalie Kallio, Nazeem Muhajarine

**Affiliations:** 1Saskatchewan Population Health and Evaluation Research Unit (SPHERU), University of Saskatchewan, 104 Clinic Place, Saskatoon, SK S7N 2Z4, Canada; alib9786@gmail.com (A.B.); daniel.adeyinka@usask.ca (D.A.A.); natalie.kallio@usask.ca (N.K.); 2Department of Community Health and Epidemiology, College of Medicine, University of Saskatchewan, 107 Wiggins Rd, Saskatoon, SK S7N 5E5, Canada; 3Canadian Hub for Applied and Social Research (CHASR), University of Saskatchewan, 9 Campus Drive, Saskatoon, SK S7N 5A5, Canada; jessica.mccutcheon@usask.ca

**Keywords:** COVID-19, dual-behavior, vaccine acceptance, vaccine refusal, consistent mask wearing, Saskatchewan

## Abstract

While the dual behavior of consistent mask wearing and vaccine acceptance represents an effective method of protecting oneself and others from COVID-19, research has yet to directly examine its predictors. A total of 3347 responses from a pooled cross-sectional survey of adults living in Saskatchewan, Canada, were analyzed using a multinomial logistic regression model. The outcome variable was the combined behavior of mask-wearing and vaccine intention in four combinations, while covariates consisted of socio-demographic factors, risk of exposure to coronavirus, mitigating behaviors, and perceptions of COVID-19. Those who were 65 years and older, financially secure, consistently practiced social distancing and had no or very few contacts with people outside their households, were concerned about spreading the virus, and perceived they would be seriously sick if infected were likely to engage in both mask wearing and vaccine acceptance, rather than one or the other, with adjusted odds ratios ranging from 2.24 to 27.54. Further, within mask wearers, these factors were associated in a graded manner with vaccine intent. By describing the characteristics of those who engage in both mask wearing and vaccine acceptance, these results offer a specific set of characteristics for public health authorities to target and, therefore, contribute to the rapidly evolving body of knowledge on protective factors for COVID-19.

## 1. Introduction

Beginning in March 2020, the province of Saskatchewan implemented several restrictions (e.g., gathering size limits, closure of non-essential businesses) to mitigate transmission of the novel coronavirus (COVID-19) [1]. These were followed by a province-wide mask mandate in November 2020, as COVID-19 cases reached a then all-time high in the province. As Saskatchewan, along with many provinces in Canada, committed to lifting these public health restrictions in mid-2021––contingent upon reaching vaccination targets––there was concern regarding individuals’ attitudes toward continuing protective behaviors, such as mask wearing and vaccination. In the first quarter of 2022, as the Omicron-variant-fueled fifth wave of the COVID-19 pandemic is sweeping across Canada, understanding the characteristics of those who engage in these protective behaviors is crucial to suppressing the current wave and preventing future waves of the pandemic. 

Researchers have highlighted the combined effect of vaccination and non-pharmaceutical public health interventions––particularly the wearing of nonmedical masks to reduce transmission––as the most promising methods of protecting oneself from COVID-19 [2,3]. The currently available vaccines are effective at increasing immunity to the SARS-CoV-2 virus and, thus, offer a longer-term solution to the pandemic through herd immunity. In Saskatchewan, however, vaccine hesitancy, defined as a “delay in acceptance or refusal of vaccines despite availability of services” [4] (p. 59), and vaccine refusal continue to pose a significant roadblock to mass immunization efforts. By early May 2021, the prevalence of vaccine hesitancy within Saskatchewan was found to be around 5% and refusal to be around 11%, posing a challenge to attaining herd immunity [5]. Commonly reported reasons for COVID-19 vaccine refusal in the literature include anxiety about vaccine side effects, lack of knowledge about vaccine effectiveness, and mistrust of the vaccine approval process [6,7]. Similarly, self-reported mask wearing was observed at 83.2% in Saskatchewan during June 2020 [8]. Commonly reported reasons associated with not wearing a mask include living in a rural setting, a lower perceived threat of COVID-19, and a lack of concern of being infected [9,10]. 

Several researchers have examined the link between mask wearing and vaccine attitudes. From a cross-sectional survey of 1056 U.S. respondents, Latkin et al. observed individuals who reported vaccine refusal were less likely to wear masks than those who reported vaccine acceptance [11]. Conversely, Rane et al. also found participants who reported wearing a mask in the month prior had substantially lower odds of being vaccine hesitant and vaccine resistant [12]. Abedin et al. reported a similar result where the majority of individuals who masked consistently also intended to vaccinate. Therefore, there is some evidence to suggest frequent mask wearing is associated with vaccine acceptance [13]. Yet, within these data, there are large differences in vaccine intention among those who mask consistently, as well as among those who do not. Not much is empirically known about the proportion of individuals who, despite reporting masking consistently, display vaccine hesitance or refusal, as no study has directly examined these behaviors as a combined outcome. By extension, the characteristics associated with those who engage with one behavior but not the other, versus who engage in both, are unknown. Given the high level of social compliance required for effective COVID-19 responses, these discrepancies in characteristics are worthy of attention when seeking to understand how to drive greater vaccine uptake and mask-wearing compliance. Behavioral correlates of vaccine acceptance, for instance, may not be associated with mask wearing for a significant amount of the population. One way to approach this issue is to treat mask wearing and vaccine acceptance as a combined, dual behavior, allowing one to examine which correlates are associated with different levels of each behavior.

Given the paucity of studies that examined the associated factors for the dual behavior of vaccine intention and mask wearing in Canada, data are needed to understand what the correlates are of engaging in the dual behavior of mask wearing and vaccine acceptance. Considering this, the current study asked whether individuals who engage in protective behaviors (e.g., social distancing, reducing contact) are more likely to both consistently wear a mask and receive a vaccine. Moreover, it sought to identify differences in characteristics among those who display differing levels of mask wearing and vaccine acceptance.

## 2. Materials and Methods

This was a pooled cross-sectional study that included responses from 3385 adults (18 years or older) residing in Saskatchewan. The data analyzed came from a broader parent study that collected behavioral, perceptual, and place-based data on COVID-19 beginning in May 2020 (Social Contours and COVID-19 Study) [5]. Data included in this study were collected between 1 January 2021 and 3 May 2021, and incorporated responses on vaccine intention after the mass vaccination campaign began in Saskatchewan on 15 December 2020, as vaccine intentions have been shown to vary depending on the availability of vaccines [14]. Mask-wearing mandates in indoor spaces and/or when unable to physically distance were also in place throughout this time period. This hybrid sample included participants from an online panel of Saskatchewan adults (i.e., community panel), originally enrolled through a probability sampling of landlines and mobile lines accessed through random digit dialing and volunteer participants recruited monthly via an online survey platform, managed by the Canadian Hub for Applied and Social Research, who collected the data, at the University of Saskatchewan, Canada. The sample size was estimated to achieve a ±1.69% margin of error, and samples were weighted using the 2016 census in terms of age, gender, and location of residence of the Saskatchewan adult population. The study protocol was reviewed and approved by the University of Saskatchewan Research Ethics Board (Beh-1971). The Social Contours study was conducted in accordance with the 2018 Tri-Council Policy Statement for the Ethical Conduct for the Research involving Humans (article 2.5). 

The outcome of interest in the present study was the dual behavior of mask wearing and COVID-19 vaccine intentions among Saskatchewan residents. Mask-wearing compliance was assessed by asking respondents: “In the last 7 days when you were inside a building (other than your home), how often did you wear a face mask (either medical grade or homemade)?” Responses for mask-wearing compliance are on a five-point frequency scale ranging from “all of the time” to “none of the time.” Vaccination intent was assessed through the question: “When a vaccine is offered to you, will you get it?” with the four response options of “yes,” “no,” “I don’t know yet,” and “I have already been vaccinated.” Responses to these questions were combined into four categories: (1) mask wearing little/none of the time and vaccine refusal or hesitant; (2) mask wearing all/most/some of the time and vaccine hesitant; (3) mask wearing all/most/some of the time and vaccine refusal; (4) mask wearing all/most/some of the time and vaccine acceptance/already received. These categories allow for a comparison of independent variables along increasingly protective levels, 1 through 4 defined above, of the behaviors against COVID-19, wherein the referent category, the first, is the least protective. Upon categorizing participants into one of these four categories, responses from 38 participants fell outside the categories and were, therefore, discarded from the analysis, resulting in a total of 3347 responses analyzed.

A total of 20 independent variables were included in the analysis based on a priori importance to the outcome [15,16,17,18,19]. These variables were layered in four broad domains: socio-demographic factors, one’s risk of exposure to coronavirus, mitigating behaviors, and perceptions of COVID-19 (see Appendix A). Potential confounding variables––such as gender, age, location of residence, and perceived financial security––were included in the analysis to adjust the potential influence of the correlates on the outcome [20].

We modeled the relationship between the dual behavior of mask wearing and vaccine intentions and all relevant independent variables using a multinomial logistic regression approach, as our outcome variable had four categories. All predictors were coded, such that the least protective category of a variable (e.g., “social distancing: little/none of the time”) acted as the referent category, with more protective categories being compared to it, such that the calculated odds ratios indicated increased protectiveness against COVID-19. To assess multicollinearity among the candidate variables, the mean variance inflation factor (VIF) was computed. With a mean VIF of 1.05, multicollinearity was not found to be a threat to internal validity. Using a backward selection approach, we initially fitted a full (saturated) multivariate multinomial logistic regression model, which adjusted for all covariates and determined the main effects of these correlates on the outcome. To avoid over, or under, fitting, the Akaike information criterion (AIC) was used to select candidate variables for retention in the parsimonious model. Model performances were assessed with adjusted R-squared, AIC, and log likelihood. Adjusted odds ratios (aOR) and 95% confidence intervals (CI) were used to estimate the strength of association. The statistical significance level of association was set at two-tailed *p*-value < 0.05. The multinomial regression model was implemented in R version 4.10 [21].

## 3. Results

The descriptive characteristics of the respondents are shown in Table 1. The average age of respondents was 56.68 years (SD: 14.06); 75.87% were women, 93.65% were born in Canada, and 73.38% had at least a technical diploma or certificate. Overall, 2.63% of respondents reported wearing a mask little/none of the time and being vaccine refusal/hesitant; 8.19% reported wearing a mask all/most/some of the time and being vaccine hesitant; 6.12% reported wearing a mask all/most/some of the time and vaccine refusal; and 83.06% reported wearing a mask all/most/some of the time and vaccine acceptance.

### Correlates of the Dual Behavior

The final model showed 50.7% of the variance in the dual behavior was explained by age, financial status, number of people in contact with outside of the household without a mask on, frequency of social distancing, self-rated personal health, perception of community adherence to public health protocols, concern about spreading the virus, perceived likelihood of being infected, and perceived severity of symptoms if infected (see Table 2).

The overarching trend observed for the majority of the above correlates was that as levels of the dual behavior outcome moved from vaccine refusal to hesitance to acceptance, its association with the correlates became progressively greater. For example, compared to those who reported they would be asymptomatic or develop minor symptoms, individuals who felt that they would either develop mild–moderate or severe symptoms/die if infected with coronavirus (all one category) were 1.59 times more likely to wear a mask consistently but refuse to be vaccinated (aOR: 1.59; 95% CI: 1.08–3.13), over twice as likely to wear a mask consistently and be vaccine hesitant (aOR: 2.14; 95% CI: 1.06–4.36), and over four times as likely to wear a mask consistently and also be vaccine acceptant (aOR: 4.42; 95% CI: 2.26–8.66).

This trend was also supported on each level of the outcome within the correlates themselves. For example, among those who reported consistent mask wearing and vaccine acceptance, the odds of being in contact with zero people over the past week (aOR: 27.54; 95% CI: 8.77–36.53), which is a more protective category, was over four times greater than being in contact with 1–5 people (aOR: 6.31; 95% CI: 3.11–12.81), which is a less protective category (see Figure 1). Similar results were observed among those who reported frequent mask wearing and vaccine hesitancy (aORs: 12.40 and 4.95, respectively) and frequent mask wearing and vaccine refusal (aORs: 5.93 and 1.63, respectively).

## 4. Discussion

The present analysis served as a baseline study to identify factors associated with both mask wearing and vaccine intention among adult residents of Saskatchewan. Overall, we observed a set of protective factors that were strongly associated with those who engage in both mask wearing and vaccine acceptance, rather than one or the other. The strongest of these factors included perceiving COVID-19 as a threat, social distancing, and anticipating severe symptoms if infected.

These results are novel, to our knowledge, as they arose from a direct examination of mask wearing and vaccine acceptance as a combined, dual behavior. The present research offers empirical support for the observation that, indeed, differences exist between those who both mask consistently and accept a vaccine versus those who practice one behavior or the other. For example, several factors associated with the dual behavior in this study overlapped with those found to correlate with either mask wearing or vaccine intention alone, including older age (over 65 years), social distancing, concern about spreading the virus, self-rated personal health, and perceived likelihood of infection [22,23,24,25,26]. However, while previous, rigorous research has consistently found the perceived severity of symptoms if infected with COVID-19 to be weakly related to vaccine intentions, this factor was strongly associated with the dual behavior in our study [27,28]. This difference in results highlights a perceptual discrepancy between those who simply report vaccine acceptance and those who report consistent mask wearing and vaccine acceptance together. Discrepancies such as this underscore the importance of examining these behaviors in combination, serving to assess individuals’ overall protection against COVID-19 more accurately. Future studies may choose to build on this result and identify other behaviors, alongside perceived severity of symptoms, which are similarly strong in those who engage in the two behaviors, but weak in those who engage in them separately. Other correlates included in the present study—including financial security, the number of people in contact with outside of the household, and whether those around the individual follow public health protocol––have not been previously examined in the literature with regard to either mask wearing or COVID-19 vaccine intention. Therefore, their strong associations with those who both mask consistently and accept a vaccine is a novel finding in the present study, which can help authorities understand more holistically the demographic and behavioral characteristics of individuals who engage in both mask wearing and vaccine acceptance.

Previous research has found consistent mask wearing is highly associated with vaccine acceptance [11,12,13,17]. In this study, the second result of differences found in vaccine intention among those who wear a mask consistently further clarifies this association. Specifically, we found that among those who consistently wear a mask, engaging in increasingly protective behaviors was directly associated with increasing levels of vaccine acceptance (i.e., vaccine refusal → hesitance → acceptance). This “gradient” of association with vaccine intentions observed among mask wearers appears logical, as researchers have observed that engaging in protective behaviors is associated with less vaccine hesitance or higher vaccine acceptance [29,30]. However, these results did not consider the heterogeneity in vaccine intentions that exist among consistent mask wearers. The present study, then, adds to the emerging knowledge base on masking and vaccine intent by providing empirical support for the existence of this trend not only between those who wear a mask consistently and those who do not, but also within the former group.

This study has a few limitations that offer avenues for future research. Overall, the sample was an imperfect representation of the Saskatchewan adult population. For example, the majority of participants were female (around 76%), while 13.2% had no/some formal education. This reflects the general tendency for more educated and affluent people and women to more likely participate in online and telephone surveys, leading to an over-representation of these groups [31,32]. Similarly, the reference category of those who engaged in inconsistent mask wearing and vaccine refusal was very small (88 out of 3347 participants), which may have resulted in wider 95% confidence intervals for the odds ratios. It is noted, however, that this proportion of the sample may indeed be a close representation of those who do not mask and refuse vaccines within our context. This study was also conducted with a Saskatchewan sample, and therefore may not be generalizable to other provinces in Canada, or to other countries. Lastly, our cross-sectional methodology does not allow for an examination of causation. Despite these limitations, the novel approach of examining masking and vaccination intent as a dual behavior provides a nuanced perspective and adds important results to the literature.

## 5. Conclusions

This study conducted on Saskatchewan adults identified several protective behaviors that were associated with both consistent mask wearing and vaccine acceptance, while observing that within those who masked consistently, individuals engaged in more of these protective behaviors had a greater likelihood of vaccine acceptance as well. These findings firstly highlight perceptual differences between those who engage in the dual behavior, rather than either masking or vaccine acceptance. Secondly, they strengthen the notion in the literature that individuals who exhibit concern for their well-being, follow public health orders, and perceive COVID-19 as a threat, to name a few correlates, are also more likely to wear a mask and accept a vaccine as a means of ending the pandemic. As such, the results add to the rapidly evolving body of knowledge on protective factors for COVID-19 by highlighting which behaviors are more likely to engage individuals in both mask wearing and vaccine acceptance, rather than one or the other. The findings may be of interest to public health authorities to direct resources toward targeting a set of behaviors for the greatest effect in health promotion campaigns by focusing on, for example, the perceived threat of symptoms if infected with COVID-19.

## Figures and Tables

**Figure 1 ijerph-19-03202-f001:**
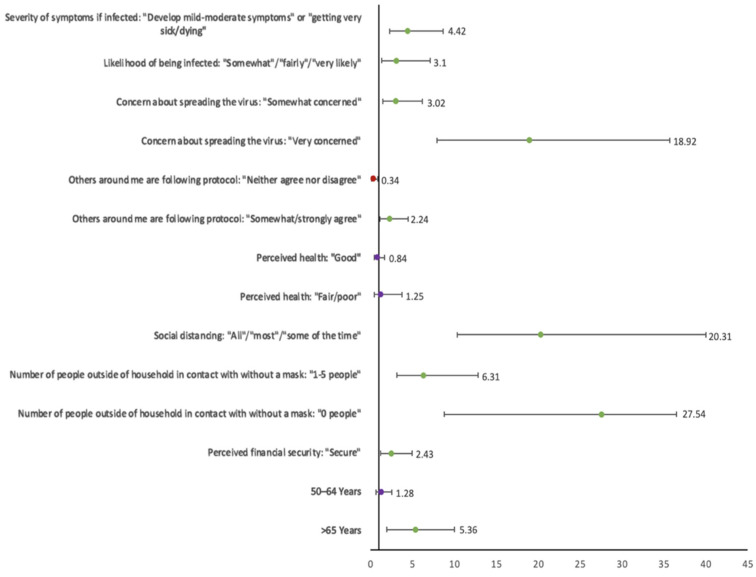
The odds ratios of covariates associated with the dual behavior, consistent mask wearing and vaccine acceptance. Green = protective effect; red = negative effect; purple = not statistically significant.

**Table 1 ijerph-19-03202-t001:** Demographic variables by category of the outcome, Saskatchewan, January–May 2021.

Demographic Variable	Mask Wearing little/None of the Time and Vaccine Refusal/Hesitant *n* (%)	Mask Wearing All/Most/Some of the Time and Vaccine Refusal *n* (%)	Mask Wearing All/Most/Some of the Time and Vaccine Hesitant *n* (%)	Mask Wearing All/Most/Some of the Time and Vaccine Acceptance *n* (%)
Overall (N = 3347)	88 (2.63)	205 (6.12)	274 (8.19)	2780 (83.06)
**Age (years) (N = 3347)**				
Under 49	55 (62.5)	97 (47.32)	88 (32.12)	785 (28.24)
50–64	25 (28.41)	66 (32.20)	122 (44.53)	1010 (36.33)
65+	8 (9.09)	42 (20.49)	64 (23.36)	985 (35.43)
**Gender (N = 3330)**				
Men	31 (35.23)	59 (28.78)	60 (21.90)	660 (23.74)
Women	51 (57.95)	140 (68.29)	204 (74.45)	2095 (75.36)
**Education (N = 3347)**				
Advanced/Professional degree	12 (13.64)	23 (11.22)	32 (11.68)	794 (28.56)
Technical diploma/certificate	43 (48.86)	99 (48.29)	143 (52.19)	1320 (47.48)
Some college or university	10 (11.36)	40 (19.51)	47 (17.15)	341 (12.27)
No/some formal education/ completed secondary	23 (26.13)	43 (20.98)	52 (18.25)	325 (11.69)
**Place of Residence (N = 3256)**				
Urban	41 (46.6)	103 (20.24)	98 (35.77)	764 (27.48)
Mid-size	16 (18.18)	29 (14.15)	47 (17.15)	350 (12.59)
Regina	12 (13.64)	32 (15.61)	50 (18.25)	633 (22.77)
Saskatoon	9 (10.23)	23 (11.22)	70 (25.55)	979 (35.22)
**Indigenous status (N = 3347)**				
Yes	7 (7.95)	6 (2.93)	11 (4.01)	73 (2.63)
No	81 (92.05)	199 (97.07)	263 (95.99)	2707 (97.37)
**Employment status (N = 3347)**				
Employed	58 (65.91)	129 (62.93)	140 (51.09)	1438 (51.73)
Unemployed/retired	30 (34.09)	76 (37.07)	134 (48.91)	1342 (48.27)

**Table 2 ijerph-19-03202-t002:** Adjusted odds ratios for the dual behavior of mask wearing and vaccine intent by correlates (*n* = 3347).

	Adjusted Odds Ratio (95% Confidence Interval)		
Variable	Mask Wearing All/Most/Some of the Time and Vaccine Refusal	Mask Wearing All/Most/Some of the Time and Vaccine Hesitant	Mask Wearing All/Most/Some of the Time and Vaccine Acceptance
**Age Groups** **(ref: below 49)**			
50–64	0.88 (0.44–1.79)	1.73 (0.84–3.56)	1.28 (0.64–2.53)
65+	2.77 (1.10–7.80) *	4.48 (1.55–13.01) **	5.36 (1.92–10.01) **
**Financially Secure (ref: Insecure)**			
Secure	0.78 (0.37–1.51)	0.77 (0.37–1.60)	2.43 (1.19–4.94) *
**Contact with people outside of the household without a mask (ref: more than 5 people)**			
1–5 people	1.63 (0.80–3.31)	4.95 (2.28–10.80) ***	6.31 (3.11–12.81) ***
0 people	5.93 (1.87–8.90) **	12.40 (3.74–20.16) ***	27.54 (8.77–36.53) ***
**Social distancing (ref: little/none of the time)**			
All/most/some of the time	5.91 (3.04–11.53) ***	11.47 (5.41–24.56) ***	20.31 (10.32–40.03) ***
**Health (ref: very good/excellent)**			
Good	0.46 (0.23–0.91) *	0.81 (0.40–1.62)	0.84 (0.43–1.63)
Fair/poor	0.54 (0.17–1.69)	0.94 (0.30–2.93)	1.25 (0.42–3.71)
**Others around me are following public health protocol (ref: somewhat/strongly disagree)**			
Neither agree nor disagree	0.62 (0.26–1.51)	0.51 (0.19–1.39)	0.34 (0.14–0.85)
Somewhat/strongly agree	2.68 (1.32–5.44) **	2.76 (1.33–5.72) **	2.24 (1.13–4.48) *
**Concern about spreading virus (ref: slightly/hardly concerned)**			
Somewhat concerned	1.10 (0.54–2.24)	2.52 (1.17–5.45) *	3.02 (1.47–6.19) **
Very concerned	1.59 (0.64–3.93)	5.83 (2.32–14.69) ***	18.92 (7.90–35.67) ***
**Perceived likelihood of being infected (ref: not at all likely)**			
Develop mild–moderate symptoms or getting very sick/dying	2.44 (1.06–5.60) *	2.44 (1.03–5.80) *	3.10 (1.35–7.11) **

Note. McFadden Pseudo R-Squared = 0.5053; Akaike information criterion = 3028.8. The adjusted odds ratios represent the likelihood of mask wearing and vaccine intention as compared to the reference category of “mask wearing little/none of the time and vaccine hesitant or refusal”.* ≤0.05. ** ≤0.01. *** ≤0.001.

## Data Availability

The use of the data presented in this study is governed by the University of Saskatchewan Data Management Policy, which legally prohibits us from sharing these data with outside investigators without official approval (https://policies.usask.ca/policies/operations-and-general-administration/data-management.php#AuthorizationandApproval; accessed on 8 March 2022). As required by the University of Saskatchewan Data Management Policy, the dataset for this study is available from the corresponding author or the university’s ethics office (ethics.office@usask.ca) on specific request.

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
