# Peer review of "Characteristics Associated with the Dual Behavior of Mask Wearing and Vaccine Acceptance: A Pooled Cross-Sectional Study among Adults in Saskatchewan"

_ijerph, 2022, doi:10.3390/ijerph19063202_

Round 1

Reviewer 1 Report

I would like to congratulate the authors for this very interesting study on the dual behavior of mask-wearing and vaccine acceptance.

The objective is well presented, the statistical methods and results are clear, the limitations well described.

I only have a few remarks:

1) The number of subject in the study

In the abstract: 3.385 responses ; in the methods: 3335 adults ; in the results: Overall N = 3347.
It's not clear to me why the number is not the same.

2) Excluded participants

A vaccinated person who never wears a mask (none of the time) doesn't fit in any of the 4 categories ; it might be very rare but when reading the paper I wonder if this scenario happened in the data and if some participants were excluded.

3) An initial chi-squared test revealed

Line 122: An initial chi-squared ...
I found this paragraph a bit surprising because this statistical result is presented before the methods and it reflects more the excitement of the researcher at the beginning of the study rather than a result of the study

Author Response

Reviewer 1: Minor Comments

  1. The number of subjects in the study: In the abstract: 3.385 responses; in the methods: 3335 adults; in the results: Overall N = 3347. It's not clear to me why the number is not the same.

Response: The original sample included 3,385 responses; however, our final model included 3,347 since 38 observations which did not fit into any 4 categories of the outcome were dropped. The 3,335 in the methods was a typo.

Changes made: Page 3, lines 125-127: “Upon categorizing participants into one of these four categories, responses from 38 participants fell outside the categories and were, therefore, discarded from the analysis, resulting in a total of 3,347 responses analyzed”

  1. Excluded participants: a vaccinated person who never wears a mask (none of the time) doesn't fit in any of the 4 categories; it might be very rare but when reading the paper. I wonder if this scenario happened in the data and if some participants were excluded.

Response: See above comment.

  1. An initial chi-squared test revealed: Line 122: An initial chi-squared. I found this paragraph a bit surprising because this statistical result is presented before the methods and it reflects more the excitement of the researcher at the beginning of the study rather than a result of the study

Response: This line has been deleted from the manuscript as it added little value to the results.

Reviewer 2 Report

In this manuscript authors evaluated if individuals who engage in protective behaviours (e.g., social distancing, reducing contact) were more likely to both consistently wear a mask and receive a vaccine. Moreover, the manuscript sought to identify differences in characteristics among those who display differing levels of mask-wearing and vaccine acceptance. 

The argument of the manuscript as described adds little to literature when states that who engage in protective behaviours is more likely to both consistently wear a mask and receive a vaccine. It is more interesting the part that evaluates protective factors for COVID-19 by highlighting which behaviours are more likely to engage individuals in both mask-wearing and vaccine acceptance, rather than one or the other, however, the introduction and discussion section are poor of contents on the implications that this study could have on national policies and what the paper adds to scientific literature. It is necessary that the authors investigate these aspects in depth, highlighting the areas in which these results could have healthcare implications. 

The Major Essential Revisions include: 

In the method section, data collection tools have to be more described. It could be useful a table with the questions reported in the questionnaire with relative answer options. It is also necessary to explain who conducted the research 

In the results section It is necessary to report the standard deviation with the average age and not the interquartile range 

The discussion section does not provide sufficient information and in-depth discussion. It does not contain new results that significantly advance the research field. Could you expand a bit more on how your study differed/added to literature? It is necessary to discuss more clearly both limitations and strengths in two different subparagraphs. 

Please, rewrite better the discussion making comparisons with more scientific references. 

Discuss potential limitations of the study within a specific paragraph, taking into account critical points, potential bias or imprecision. 

Author Response

Reviewer 2: Major comments

  1. In the method section, data collection tools have to be more described. It could be useful a table with the questions reported in the questionnaire with relative answer options. It is also necessary to explain who conducted the research 

Response: We have added the key and relevant measures included in the study in Appendix A. In addition, we state the parent study (the Social Contours and COVID-19 Study) and give a link to this study where more information can be viewed (lines 94-95). We also had clearly stated who had conducted the study in Author Contributions, but prior to that in page 3, lines 100-105: “This hybrid sample included participants from an online panel of Saskatchewan adults (i.e., Community Panel), originally enrolled through a probability sampling of landlines and mobile lines accessed through random digit dialing, and volunteer participants recruited monthly via an online survey platform, managed by the Canadian Hub for Applied and Social Research, who collected the data, at the University of Saskatchewan, Canada.

  1. In the results section It is necessary to report the standard deviation with the average age and not the interquartile range 

Response: The IQR has been replaced with the SD (page 4; line 156).

  1. The discussion section does not provide sufficient information and in-depth discussion. It does not contain new results that significantly advance the research field. Could you expand a bit more on how your study differed/added to literature? It is necessary to discuss more clearly both limitations and strengths in two different subparagraphs. 

Response: The two main findings of this study were: (1) that a strong set of protective factors were associated with those who engaged in both mask-wearing and vaccine acceptance, rather than one or the other. These factors differed from those previous findings which have reported for a single behaviour, either masking or vaccine acceptance. Therefore, this finding is a novel result which allows for a better understanding of discrepancies between individuals who engage in the dual behaviour (which is ideal for reduced transmission) and those who do not; and (2) differences found in vaccine intention (accept, hesitate and refuse) among those who wear a mask consistently. Previous research has treated consistent mask-wearers as a homogenous group with regards to vaccine intention, and we not only showed this is not the case, but also found that similar trends occur within those who mask consistently as between them and those who do not.

Changes made: Much of the discussion section has been updated to reflect the above-stated important and useful findings.

Page 8, lines 202-206: “Overall, we observed a set of protective factors that were strongly associated with those who engage in both mask-wearing and vaccine acceptance, rather than one or the other. The strongest of these factors included perceiving COVID-19 as a threat, social distancing, and anticipating severe symptoms if infected.”

Page 8, lines 207-211: “These results are novel, to our knowledge, as they arose from a direct examination of mask-wearing and vaccine acceptance as a combined, dual behaviour. The present research offers empirical support for the casual observation that indeed differences exist between those who both mask consistently and accept a vaccine versus those who practice one behaviour or the other”

Page 8, lines 223-226: “Future studies may choose to build on this result and identify other behaviours, alongside perceived severity of symptoms, which are similarly strong in those who engage in the two behaviours, but weak in those who engage in them separately.”

Page 8, lines 230-234: “Therefore, their strong associations with those who both mask consistently and accept a vaccine is a novel finding, in the present study, which can help authorities more holistically understand the demographic and behavioural characteristics of individuals who engage in both mask-wearing and vaccine acceptance.”

Page 8, lines 244-248: “However, these results did not consider the heterogeneity in vaccine intentions that exist amongst consistent mask-wearers. The present study, then, adds to the emerging knowledge base on masking and vaccine intent by providing empirical support for the existence of this trend not only between those who wear a mask consistently and those who do not, but also within the former group.”

  1. Discuss potential limitations of the study within a specific paragraph, taking into account critical points, potential bias or imprecision. 

Response: Reviewer 1 raised similar point, and we have addressed this. Please see our response to this comment above.

Reviewer 3 Report

Authors mentioned the importance of finding this study in conclusion. 
“The findings may be of interest to public health authorities to direct resources towards targeting a set of behaviours for the greatest effect in health promotion campaigns”.

And I am also agreeing with authors and public health authorities are interested in this finding. 

Author Response

Reviewer 3: No comments

Round 2

Reviewer 2 Report

The authors replied and edited the manuscript as required